# High-Efficiency Copper Removal by Nitrogen Plasma-Assisted Picosecond Laser Processing

**DOI:** 10.3390/mi13091492

**Published:** 2022-09-08

**Authors:** Yunfan Li, Xuanqi Guo, Shuai Wang, Shizhuo Zhang, Yilin Zhao, Dingyi Guo, Chen Zhang, Sheng Liu, Gary J. Cheng, Feng Liu

**Affiliations:** 1School of Power and Mechanical Engineering, Wuhan University, Wuhan 430072, China; 2Institute of Technological Sciences, Wuhan University, Wuhan 430072, China; 3School of Industrial Engineering, Purdue University, West Lafayette, IN 47906, USA

**Keywords:** Cu removal, ps-laser, nitrogen plasma, high-efficiency

## Abstract

Copper (Cu) removal efficiency is a key parameter in the processing of Cu-based electronic devices. Herein, a nitrogen plasma-assisted picosecond (ps) laser process for Cu removal is presented. Based on the cleaning and activation effect of nitrogen plasma on the surface of Cu film in ps-laser ablation, the removal efficiency can be significantly improved. Theoretically, the interaction mechanism between Cu and the ps-laser under the action of the plasma flow field is investigated by the dual temperature model (TTM) and finite element analysis (FEA). Meanwhile, the experimental results show that the angle of the plasma flow significantly affects the laser ablation of Cu. Small-angle plasma helps to improve the ps-laser processing precision of Cu, while large-angle plasma can effectively improve the ps-laser processing efficiency of Cu. Under the laser fluence of 2.69 J/cm^2^, the removal depth of the Cu film by a 30° plasma-assisted ps-laser is 148% higher than that by the non-plasma-assisted ps-laser, which indicates the application potential of nitrogen plasma in improving the laser ablation process.

## 1. Introduction

Since copper (Cu) is a metal with excellent electrical conductivity, high resistance to electromigration and low thermal sensitivity, it is widely used in electronic devices [1], high-temperature sensing [2] and mechanical transmission [3]. Therefore, the realization of an efficient Cu removal process is crucial in facilitating the industrial application of Cu. Existing methods for Cu removal in electronic devices mainly include photolithography-wet etching [4,5,6,7,8,9] and mechanical polishing [10,11,12,13,14,15]. The photolithography-wet etching method can achieve nanoscale machining accuracy and nm/min removal rate, but there are some limitations such as complexity, high cost, time consumption and environmental unfriendliness [16,17,18,19]. The mechanical polishing method is simple and high in efficiency, but it is easy to cause mechanical damage due to contact processing, and it is difficult to obtain nanoscale machining accuracy [20,21,22,23]. Recently, as an advanced and non-contact manufacturing technology, laser processing has been widely used in the fields of automobile, electronics, and machinery manufacturing. The accuracy of laser processing is closely related to the pulse width of the laser used. Laser processing based on the ultrafast laser can achieve nanoscale removal accuracy, μm/min removal rate and low mechanical damage [24,25], which has great application potential in Cu processing of electronic devices.

Therefore, laser processing has been widely used in Cu micromachining, such as surface modification [26,27,28], nanostructure fabrication [29,30,31], induction of plasma [32,33,34], high-precision removal [35,36,37,38,39] and selective removal [40,41,42]. The machining accuracy of a long pulse laser is limited due to the obvious thermal effect. The picosecond laser is an ultrafast laser with a low thermal effect and high processing precision, which is very suitable for Cu processing of electronic devices. Processing efficiency and processing accuracy are the focus of ps-laser processing. At present, there is still large room for improvement in the efficiency and precision of laser processing. Recently, some studies have been conducted to improve laser processing with plasma assistance. For example, plasma-assisted laser processing was used to create microtextures on carbide surfaces to enhance the adhesion of PVD coatings [43]. High-quality microgrooves were fabricated by combining laser-induced plasma with magnetic fields [44]. The removal efficiency of laser ablation of titanium was improved by plasma assistance [45]. However, the coupling mechanism of incident plasma and the effect of plasma angle on laser ablation have not been investigated. More importantly, studies on plasma-assisted laser processing of Cu have not been reported.

In this study, a nitrogen plasma-assisted ps-laser ablation process is proposed to achieve high-efficiency Cu removal. Theoretically, the temperature field evolution of Cu irradiated by a ps-laser coupled with a nitrogen plasma flow field is investigated by combining the dual temperature model (TTM) with the fluid–solid heat transfer model. Experimentally, the effect of the plasma action angle on nitrogen plasma-assisted ps-laser removal of Cu is investigated. The theoretical and experimental results show that plasma action angle has a significant effect on the removal of Cu by ps-laser. A reasonable plasma action angle can effectively improve the removal efficiency of the ps-laser due to the cleaning and activation effect of plasma flow.

## 2. Experiment

### 2.1. Materials

High-purity silicon wafers (Lijing Photonics Co., Ltd., Zhejiang, China) with (100) crystal orientation and a thickness of 400 μm are used. The silicon wafers are ultrasonically cleaned using acetone, ethanol and deionized water in sequence, and then a 1.5 μm thick Cu film is deposited on the wafers by magnetron sputtering.

### 2.2. Experimental Process

The processing diagram of nitrogen plasma-assisted ps-laser removal of Cu is shown in Figure 1a. The wavelength, pulse width, maximum repetition frequency and maximum output power of the ps-laser (TruMicro 5000, TRUMPF Scientific Lasers Co., Ltd., Unterföhring, Germany) are 532 nm, 12 ps, 600 kHz and 100 W, respectively. During the process of Cu removal, the repetition frequency and defocus of the ps-laser are set to 600 kHz and 0 μm, and the scanning speed (*v*), scanning pitch (Δ*d*) and spot diameter are 10,000 mm/s, 10 μm and 10 μm, respectively. The pulse laser is focused on the sample surface by a galvanometer and scanned on the sample surface according to the path set by the computer.

The plasma generator (SPA-2600, SINDIN Co., Ltd., Dongguan, China) consists of a high voltage excitation power supply, a spray gun and a control system. Under the excitation of a high voltage power supply, the compressed nitrogen with a pressure of 0.3 MPa is ionized into plasma, and then the plasma is sprayed by the spray gun to act on the copper film. The voltage and frequency of the excitation power supply are 220 V and 40 kHz, respectively. The generating power, diameter and temperature of nitrogen plasma flow are 180 W, 5 mm and 70°, respectively. During the ps-laser irradiation of Cu film, the plasma spray gun is installed on the rack, and the ps-laser ablates Cu in the plasma flow action region.

### 2.3. Characterization

The surface micromorphology and elemental composition of the Cu film are observed by scanning electron microscopy (Zeiss Sigma, Carl Zeiss AG Co., Ltd., Oberkochen, Germany). The 3D morphology of the Cu film is characterized by an optical profiler (NewView 9000, Zygo, CT, USA) to obtain the surface roughness and removal depth. The crystalline phase of the samples is characterized by an X-ray diffractometer (XRD, XPert Pro, PANalytical B.V., Almelo, The Netherlands).

## 3. Theoretical Analysis and Simulation

### 3.1. Mechanism Analysis

Figure 1b shows the simultaneous action of the nitrogen plasma and the ps-laser on the Cu film. When the ps-laser irradiates Cu film, the lattice temperature increases rapidly after a brief period of non-thermal equilibrium. When the temperature of the lattice exceeds the evaporation temperature of Cu, the direct conversion of Cu from a solid to a gaseous state occurs, which is consistent with refs. [46,47]. The influence of plasma on the laser ablation process is related to angle *a* between the plasma flow direction and the sample surface. When angle *a* is small, the nitrogen plasma flow forms a protective layer on the Cu film [48,49] and weakens the ps-laser’s ablation effect on the Cu film. As a result, the removal depth of Cu ablated by the ps-laser with plasma flow is less than that of the ps-laser without plasma flow. As the angle increases, a lot of active particles such as ions, excited state molecules and free radicals in the nitrogen plasma flow act on the Cu film surface [50,51], which play the role of cleaning the deposited melt and activating the surface. Thus, the effect of ps-laser ablation is enhanced, and the removal depth of the Cu film is increased.

### 3.2. Numerical Model

The numerical model in this work involves the coupling of the plasma flow field and ps-laser ablation. The two-temperature model (TTM) is used to simulate the solid heat transfer process of Cu film ablated by ps-laser, and the laminar flow model is used to simulate the nitrogen plasma flow field. During the interaction of the ps-laser with the irradiated metal, the duration of the laser action is less than the time required to reach the thermal equilibrium temperature. At first, there is a large temperature difference between the electron and the lattice. Subsequently, the electron transfers heat to the lattice, and the lattice begins to heat up. The whole temperature rise process can be divided into three stages: the first stage is the absorption of photon energy by electrons, the second stage is the transfer of electron energy to the lattice and the third stage is the interaction between lattice atoms. Currently, the TTM is widely used for heat transfer in ultrafast laser irradiation processes. According to the TTM in refs. [52,53,54], the temperature–time evolution process of fs-laser–matter interaction is described by the following nonlinear differential equation.
(1)$$\begin{array}{l} C_{e}\frac{\partial T_{e}}{\partial t}=\nabla (k_{e}\nabla T_{e})-g(T_{e}-T_{l})+S(r,t)\\ C_{l}\frac{\partial T_{l}}{\partial t}=g(T_{e}-T_{l}) \end{array}$$
where *r* is the distance from the laser center, *t* is the time and *g* is the electron–phonon coupling factor. *k_e_* is the electron thermal conductivity, and *S* (*r*, *t*) represents the absorbed laser heat source. *C_e_* and *C_l_* are the electron thermal capacity and lattice heat capacity, respectively. *T_e_* and *T_l_* are the electron temperature and lattice temperature, respectively.

In this study, a Gaussian pulsed laser is used, and the heat source can be simulated by the following expression.
(2)$$S(r,t)=\frac{I_{0}}{\tau_{p}}(1-R)\alpha_{b}\exp(-\alpha_{b}z)\exp[-2{\left(\frac{r}{r_{0}}\right)}^{2}]\exp[-4\ln2{\left(\frac{t-\tau_{p}}{\tau_{p}}\right)}^{2}],$$
where *I*_0_ is the maximum value of the laser fluence, $\tau_{p\;}$ is the pulse width of the laser, $\alpha_{b}$ is the material absorption coefficient, *R* is the material reflectivity and $r_{0\;}$ is the radius of the laser beam.

Since the nitrogen ions in the nitrogen plasma do not react chemically with Cu, the nitrogen plasma flow field flowing on the surface of Cu film can be simulated by the laminar flow equation.
(3)$$\rho \frac{\partial \vec{u}}{\partial t}+\rho \left(\vec{u}\cdot \nabla \right)\vec{u}=\nabla \cdot \left[-p\vec{I}+\vec{K}\right]+\vec{F},\;\rho \nabla \cdot \vec{u}=0$$
where *ρ* is the fluid density, $\vec{I}\;$ is the unit vector, $\vec{u}\;$ is the velocity field component, $\vec{K}\;$ is the viscous force component, *p* is the gas pressure and $\vec{F}\;$ is the laser flux component.

The solid–fluid heat transfer equation is used to simulate the interaction of the nitrogen plasma flow field with the temperature field of ps-laser ablation.
(4)$$\rho C_{p}\frac{\partial T}{\partial t}+\rho C_{p}\vec{u}\cdot \nabla T+\nabla \cdot \vec{q}=Q+Q_{ted},\;\vec{q}=-k\nabla T$$
where *ρ* is the fluid density, $\vec{q}\;$ is the ablation heat flux, *C_p_* is the heat capacity of the solid, *Q* and *Q_ted_* are the heat source, *T* is the fluid temperature, $\vec{u}\;$ is the velocity field component and *k* is the thermal conductivity.

## 4. Results and Discussion

### 4.1. Simulation Results

The time-dependent evolution of lattice temperatures in Cu irradiated by a single-pulse 1.0 J/cm^2^ ps-laser with different plasma action angles is simulated, as shown in Figure 2. A two-dimensional (2D) axisymmetric Cu film with a thickness of 1 μm is modeled in COMSOL Multiphysics, and a virtual rectangular wind field area is set up above the Cu film. The edge of the rectangular notch is the entrance of the wind field. Figure 2a shows the 2D temperature distribution of the Cu film as 10 ps after being irradiated by a plasma-assisted ps-laser with a plasma flow angle of 0°. It can be observed that the temperature in the center of the ps-laser heat source is the highest, and the temperature sequentially decreases from the center to the edge, which is consistent with the characteristics of Gaussian lasers. The same results can be observed in Figure 2b,c. The comparison of the lattice temperature evolution ablated by ps-lasers with different plasma flow angles can be observed in Figure 2d. The lattice temperature exceeds the evaporation temperature after a short time interval (6 ps), resulting in the removal of Cu. The lattice temperature peaks at 14 ps and then begins to decrease, which is consistent with the pulse width of the ps-laser. Furthermore, it can be observed in Figure 2d that the peak lattice temperature of Cu film irradiated by a plasma-assisted ps-laser with a 0° plasma action angle is the lowest, and the peak temperature is positively correlated with the angle. This is because the small-angle plasma has a shielding effect on the process of ps-laser irradiating the Cu film, which weakens the heat transfer of ps-laser ablation. The Cu film under large-angle plasma-assisted ps-laser irradiation has a higher peak lattice temperature; thus, the removal depth is increased, which is consistent with the working mechanism proposed in Section 3.1.

### 4.2. Cu Removal by PS-Laser without Plasma Assistance

The surface and 3D morphologies of Cu films ablated by a ps-laser without plasma assistance are investigated, as shown in Figure 3. The scanning electron microscopy (SEM) image in Figure 3a shows the deposition of molten particles on the Cu film irradiated by a ps-laser with 2.69 J/cm^2^ fluence. As shown by the 3D profile, the surface roughness and removal depth of Cu film are 32.19 nm and 153 nm, respectively. In Figure 3b–d, more molten deposition can be observed as the laser fluence increases. Thus, the surface roughness and removal depth increase correspondingly. This is because the increased laser fluence has two effects on laser ablation. One effect is that the thermal effect of the laser is enhanced, and the other effect is that the etching effect of the laser-induced plasma [55,56] is also enhanced. Specifically, the surface roughness of the Cu films processed by ps-lasers with 3.14 J/cm^2^, 3.59 J/cm^2^ and 4.04 J/cm^2^ fluences are 34.05 nm, 45.61 nm and 62.32 nm, respectively. The surface quality decreases sequentially. The removal depths of the Cu films processed by ps-lasers with 3.14 J/cm^2^, 3.59 J/cm^2^ and 4.04 J/cm^2^ fluences are 362 nm, 474 nm and 574 nm, respectively. The removal efficiency increases sequentially.

### 4.3. Cu Removal by PS-Laser with Plasma Assistance

The morphologies of Cu films ablated by a nitrogen plasma-assisted ps-laser with 0° plasma flow angle are investigated. In Figure 4, with the increase in laser fluence, it can be observed that the ablation traces increase, the surface quality of Cu film decreases and the ablation depth of Cu film increases, which is consistent with the Cu film ablated by the ps-laser without plasma. Compared to the samples by the ps-laser in air (Figure 3), the Cu films processed by the 0° nitrogen plasma-assisted ps-laser show fewer ablation traces and better surface quality. Specifically, under the irradiation of the ps-laser with 4.04 J/cm^2^ fluence, the surface roughness of the Cu film processed in 0° nitrogen plasma flow is 52.14 nm, which is lower than 62.32 nm of the sample processed in air (Figure 3d), while the removal depth of the Cu film is 574 nm, which is lower than 505 nm of the sample processed in air (Figure 3d). The experimental results indicate that the small-angle plasma flow has a shielding effect on the laser photons, which leads to the weakening of the laser energy acting on the Cu film, thus improving the surface quality of Cu film and reducing the removal efficiency of laser ablation.

The effects of nitrogen plasma action angles of 15° and 30° on the ps-laser ablation of Cu films are also investigated, as shown in Figure 5 and Figure 6, respectively. Compared with the 0° plasma-assisted sample (Figure 4d), the surface quality of the Cu film ablated by the 15° nitrogen plasma-assisted ps-laser at 4.04 J/cm^2^ (Figure 5d) decreases, but the removal efficiency increases. Specifically, the surface roughness increases from 52.14 nm to 78.97 nm, and the removal depth increases from 505 nm to 599 nm. When the nitrogen plasma angle is 30°, the effect is more significant, the surface roughness of the Cu film increases from 78.97 nm to 85.74 nm and the removal depth increases from 599 nm to 674 nm. The experimental results show that the plasma at large angles (15°, 30°) can promote ps-laser ablation, and the promoting effect is positively correlated with the angle. This is because the larger the angle, the closer the direction of the plasma flow and the direction of the laser photons, the weaker the shielding effect of the plasma on the laser energy and the stronger the cleaning and activation effects of the plasma on the surface of the Cu film. However, it also forms many micro-pits in the action area, which increases the surface roughness.

Figure 7 shows the surface roughness and removal depth of Cu films ablated by a nitrogen plasma-assisted ps-laser with different laser fluences at different plasma action angles. As shown in the figure, the surface roughness and the ablation depth are positively correlated with the laser fluence. This is because the Cu film absorbs more energy, and the ablation is more intense under the irradiation of a ps-laser with high fluence. In Figure 7a, the surface quality of the Cu films ablated by a 0° plasma-assisted ps-laser is always better than that of the Cu films without plasma assistance due to the shielding effect of the plasma flow. However, when the angle is increased to 15° and 30°, the surface quality of the Cu films ablated by the plasma-assisted ps-laser is always lower than that of the Cu films without plasma assistance due to the promoting effect of plasma on laser ablation. In contrast, as can be observed in Figure 7b, the 15° and 30° nitrogen plasma assistance increase the removal depth of Cu film, while the 0° nitrogen plasma assistance reduces the removal depth. This is because the plasma flow at 0° is parallel to the Cu film and gathers above the Cu film to form a protective layer, thus weakening the laser ablation. At 15° and 30°, the plasma can act directly on the surface to clean the molten particles, thus promoting laser ablation. For example, at a laser fluence of 2.69 J/cm^2^, the removal depth of the Cu film ablated by a 30° plasma-assisted ps-laser is improved by 290% compared to 0°, but the surface roughness of the Cu film is increased by 56%. The removal depth of the 0° plasma-assisted Cu film was reduced by 57% compared to no plasma, but the surface roughness was reduced by 42%. The experimental results show that changing the angle of the nitrogen plasma can produce two different effects, weakening or promoting the ps-laser ablation of Cu. Therefore, the processing quality of ps-laser removal of Cu can be controlled by adjusting the plasma action angle.

### 4.4. Analysis of the Processed Cu Films

The phases of the Cu films ablated by the ps-laser are investigated. Figure 8a shows the XRD patterns of the Cu films ablated by the 2.69 J/cm^2^ ps-laser. In the figure, the peaks of CuO (110), CuO (111), CuO (113) and Cu (111) appear in the sample ablated by the ps-laser, which indicates that a small amount of Cu is oxidized to CuO during ps-laser ablation. No peak related to the nitrogen element appears in the XRD patterns, which indicates that the nitrogen plasma does not react with Cu. In addition, the pattern of Figure 8(a1) is basically the same as that of Figure 8(a2), indicating that the assistance of nitrogen plasma has little effect on the phase structure of the Cu film. Figure 8b shows the elemental compositions of the samples ablated by a 2.69 J/cm^2^ ps-laser. In the figure, the Cu film ablated by a nitrogen-assisted ps-laser contains a large amount of Cu and a small amount of C and O. This proves that the Cu film does not react with nitrogen after being ablated by a nitrogen-assisted ps-laser, which is consistent with the XRD results. The presence of small amounts of C may be caused by the ablated organic matter remaining on the surface of the Cu film. The presence of element O indicates that a small amount of copper oxide formed by the reaction between Cu and oxygen in the air is deposited on the Cu film during ablation. In addition, the oxygen content of the sample ablated in nitrogen plasma is almost the same as that of the sample ablated in air, further indicating that the assistance of nitrogen plasma has little effect on the phase structure of the Cu film.

## 5. Conclusions

In this study, a high-efficiency nitrogen plasma-assisted ps-laser ablation process was proposed for Cu removal in electronic devices. Theoretically, based on the fluid–solid heat transfer multi-physics coupling model, the mechanism of ps-laser ablation of Cu under the action of nitrogen plasma at different angles was investigated. Experimentally, the morphologies of Cu films ablated by the ps-lasers with different fluences at different plasma action angles were investigated. The surface quality and removal depth of Cu films ablated by nitrogen plasma-assisted ps-laser were different with different plasma action angles, which demonstrated the modulating effect of nitrogen plasma on the ps-laser ablation of Cu. When laser fluence was 2.69 J/cm^2^, the removal depth of Cu film ablated by 30° plasma-assisted ps-laser was 290% higher than that of the 0° plasma-assisted ps-laser and 148% higher than that of the ps-laser without plasma assistance. Therefore, the removal efficiency can be greatly improved by large-angle plasma assistance, which is beneficial in promoting the application of the laser ablation method in the processing of electronic devices.

## Figures and Tables

**Figure 1 micromachines-13-01492-f001:**
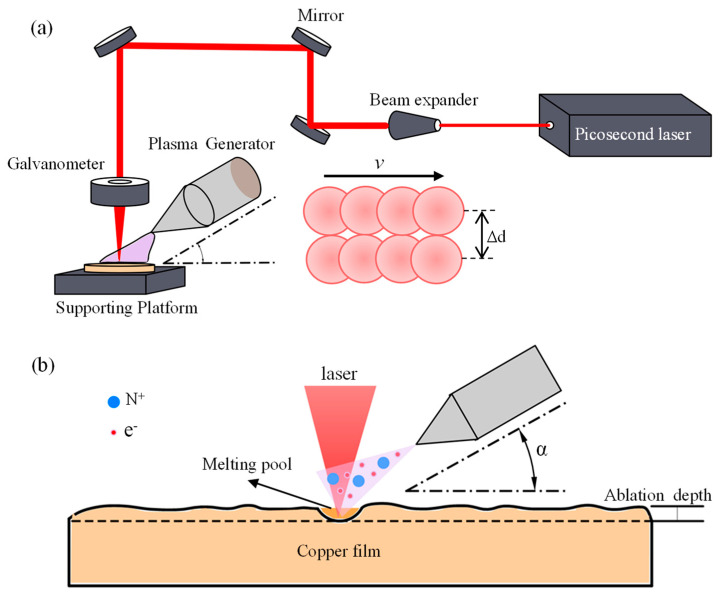
(**a**) Experimental schematic of Cu removal by nitrogen plasma−assisted ps-laser. (**b**) Mechanism of Cu removal by nitrogen plasma-assisted ps−laser.

**Figure 2 micromachines-13-01492-f002:**
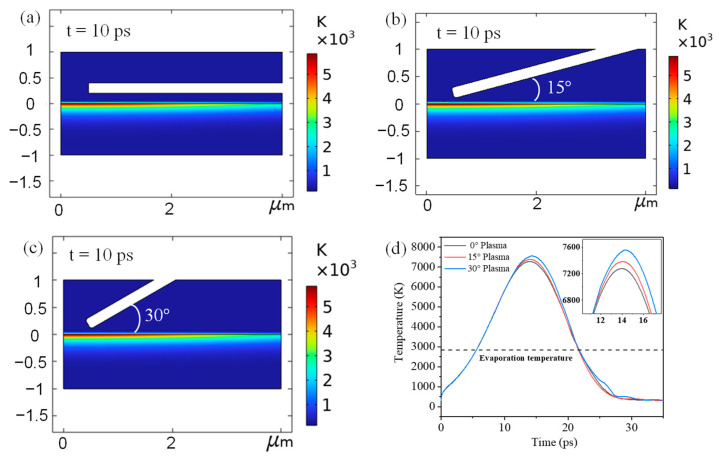
Lattice temperature distribution of Cu film at 10 ps after being irradiated by a single−pulse 1.0 J/cm^2^ ps−laser with different plasma action angles: (**a**) 0°, (**b**) 15° and (**c**) 30°. (**d**) Evolution of lattice temperatures with time in Cu irradiated by single−pulse 1.0 J/cm^2^ ps−laser with different plasma action angles.

**Figure 3 micromachines-13-01492-f003:**
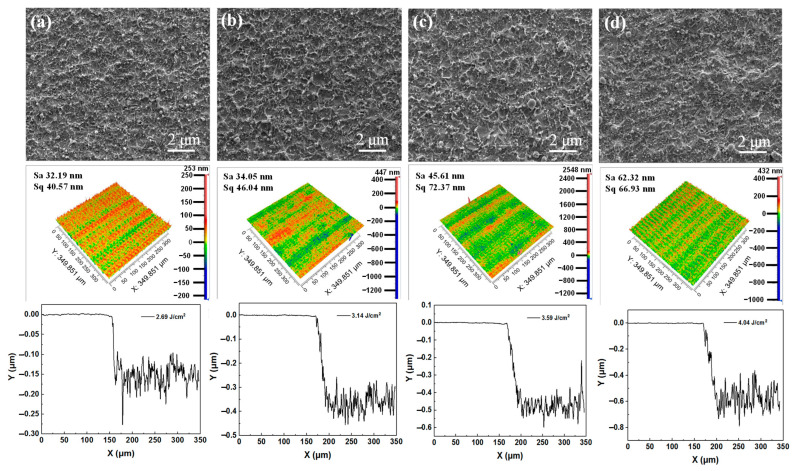
Surface morphologies and 3D morphologies of Cu films ablated by the ps−lasers at different laser fluences: (**a**) 2.69 J/cm^2^, (**b**) 3.14 J/cm^2^, (**c**) 3.59 J/cm^2^ and (**d**) 4.04 J/cm^2^.

**Figure 4 micromachines-13-01492-f004:**
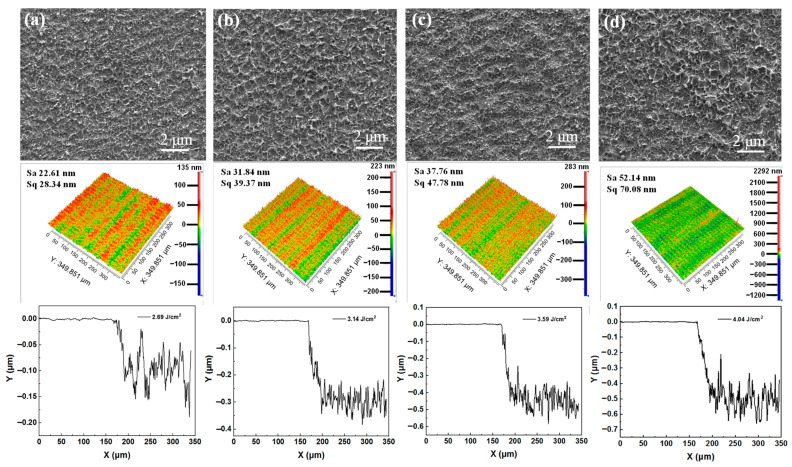
Surface morphologies and 3D morphologies of Cu films ablated by the 0° nitrogen plasma− assisted ps−lasers at different laser fluences: (**a**) 2.69 J/cm^2^, (**b**) 3.14 J/cm^2^, (**c**) 3.59 J/cm^2^ and (**d**) 4.04 J/cm^2^.

**Figure 5 micromachines-13-01492-f005:**
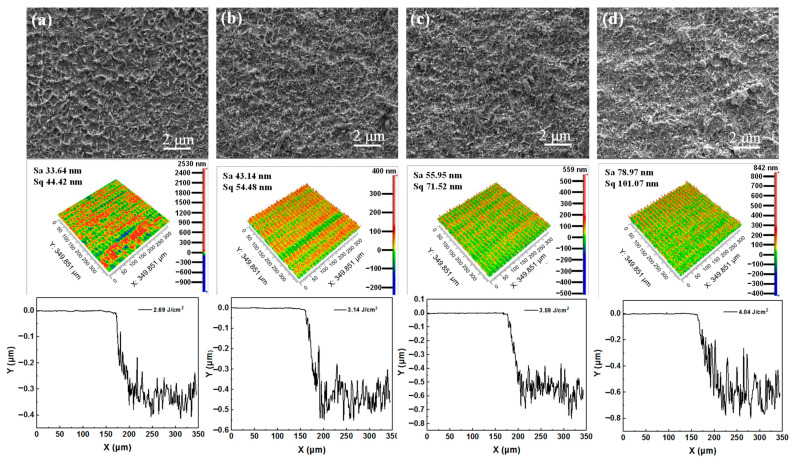
Surface morphologies and 3D morphologies of Cu films ablated by the 15° nitrogen plasma−assisted ps−lasers at different laser fluences: (**a**) 2.69 J/cm^2^, (**b**) 3.14 J/cm^2^, (**c**) 3.59 J/cm^2^ and (**d**) 4.04 J/cm^2^.

**Figure 6 micromachines-13-01492-f006:**
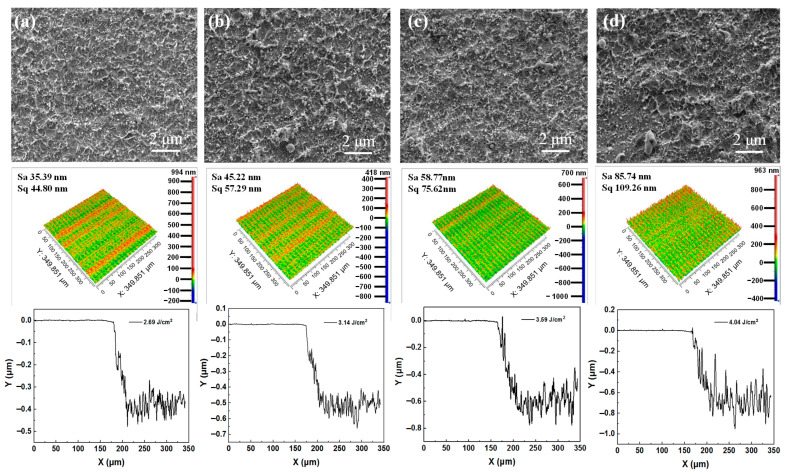
Surface morphologies and 3D morphologies of Cu films ablated by the 30° nitrogen plasma−assisted ps−lasers at different laser fluences: (**a**) 2.69 J/cm^2^, (**b**) 3.14 J/cm^2^, (**c**) 3.59 J/cm^2^ and (**d**) 4.04 J/cm^2^.

**Figure 7 micromachines-13-01492-f007:**
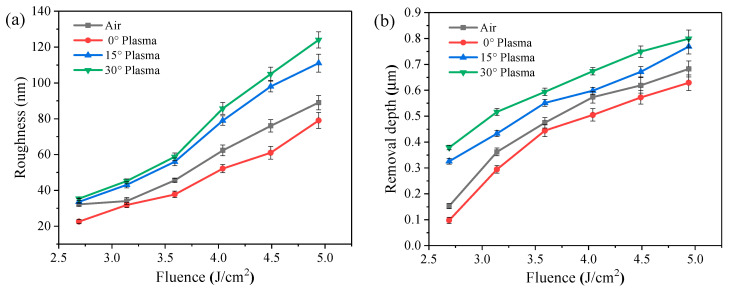
Correlation between laser fluence and processing quality of Cu films ablated by nitrogen plasma−assisted ps−lasers: (**a**) roughness and (**b**) removal depth.

**Figure 8 micromachines-13-01492-f008:**
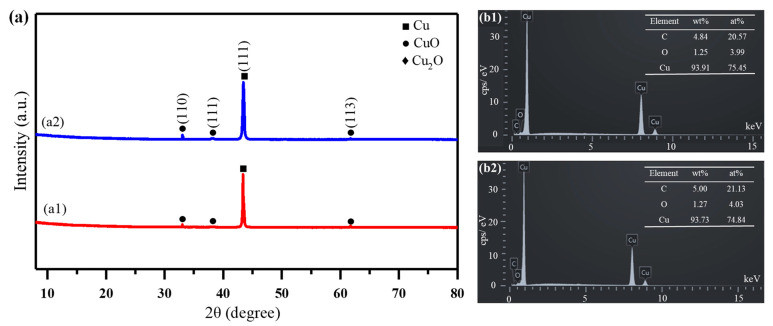
(**a**) XRD results of the Cu films ablated by a 2.69 J/cm^2^ ps−laser: (**a1**) without the plasma assistance, (**a2**) with the 30°plasma assistance. (**b**) Energy dispersive spectrometer (EDS) results of the Cu films ablated by a 2.69 J/cm^2^ ps−laser: (**b1**) without the plasma assistance and (**b2**) with the 30°plasma assistance.

## Data Availability

Not applicable.

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
