# Peer review of "High-Efficiency Copper Removal by Nitrogen Plasma-Assisted Picosecond Laser Processing"

_micromachines, 2022, doi:10.3390/mi13091492_

Round 1

Reviewer 1 Report

The authors demonstrated a beneficial study of High-efficiency Copper Removal by Nitrogen Plasma-assisted 2
Picosecond Laser in various areas. More details are needed as follows;
1- The English structure showed to be improved a lot.
I have attached a corrected file with some suggestions as a model for corrections.
2- In the introduction, the idea that laser induces copper plasma should be reviewed in detail. The ns and ps laser induce plasma of copper showed be given. Also, recent references should be included as well. For example;
A- Penttilä, R., H. Pantsar, and P. Laakso. "Picosecond laser processing–material removal rates of metals." In Proceedings of the 11th NOLAMP Conference in Laser Processing of Materials, pp. 502-512. 2007.
B- Schille, Joerg, Lutz Schneider, and Udo Loeschner. "Process optimization in high-average-power ultrashort pulse laser microfabrication: how laser process parameters influence efficiency, throughput and quality." Applied Physics A 120, no. 3 (2015): 847-855.
C- Fikry, Mohamed, Walid Tawfik, and Magdy Omar. "Measurement of the Electron Temperature in a Metallic Copper Using Ultrafast Laser-Induced Breakdown Spectroscopy." Journal of Russian Laser Research 41, no. 5 (2020): 484-490.
D- Fikry, Mohamed, Walid Tawfik, and Magdy M. Omar. "Investigation on the effects of laser parameters on the plasma profile of copper using picosecond laser induced plasma spectroscopy." Optical and Quantum Electronics 52 (2020): 249.
E- Fikry, Mohamed, Walid Tawfik, and Magdy M. Omar. " Controlling the plasma electron number density of copper metal using NIR picosecond laser-induced plasma spectroscopy." Optica Applicata 51 (3) 2021.
F- Li, Yunfan, Xuanqi Guo, Shuai Wang, Yilin Zhao, Chen Zhang, Sheng Liu, Gary J. Cheng, and Feng Liu. "High-precision and high-efficiency copper removal by femtosecond laser in mixed hydrochloric acid‑oxygen atmosphere." Journal of Manufacturing Processes 82 (2022): 152-158.
3- Many technical details should be given; the used laser focusing optics, the generated dc plasma source details, including plasma temperature and density, and how the synchronization is done between the stationary plasma source and the laser?
4- The depth profile through the SEM images is not clear. I highly recommend using the TEM images instead.

Author Response

Dear Editor and Reviewers,

Thank you for your letter and for the reviewer’s comments concerning our manuscript (micromachines-1894310) entitled “High-efficiency Copper Removal by Nitrogen Plasma-assisted Picosecond Laser”. Those comments are all valuable and very helpful for revising and improving our paper, as well as the important guiding significance to our research. According to the reviewers’ and the associate editor’s comments, we have carefully revised our manuscript, and hope that our revision is satisfactory for publication in Micromachines. In this revised version, the changes to our manuscript were all highlighted within the document by using red-colored text. The main corrections in the manuscript and the details of the responses to the comments are listed below. If there are any other questions or comments, please feel free to contact us.

Feng Liu

School of Power and Mechanical Engineering

Wuhan University

Wuhan, Hubei 430072, China

E-mail: fengliu@whu.edu.cn

2022-8-30

Responses to the reviewers’ comments:

Reviewer 1:
The authors demonstrated a beneficial study of High-efficiency Copper Removal by Nitrogen Plasma-assisted Picosecond Laser in various areas. More details are needed as follows.

Response: Thank you very much for your comments, we have revised the manuscript based on the comments and provided our point-to-point responses below.

1- The English structure should be improved a lot. I have attached a corrected file with some suggestions as a model for corrections.

Response: Thank you for your advice, we have made many modifications base on your proposal. Instead of listing these changes here, we have marked the changes in red in the revised manuscript.

2- In the introduction, the idea that laser induces copper plasma should be reviewed in detail. The ns and ps laser induce plasma of copper should be given. Also, recent references should be included as well. For example:

A- Penttilä, R., H. Pantsar, and P. Laakso. "Picosecond laser processing–material removal rates of metals." In Proceedings of the 11th NOLAMP Conference in Laser Processing of Materials, pp. 502-512. 2007.

B- Schille, Joerg, Lutz Schneider, and Udo Loeschner. "Process optimization in high-average-power ultrashort pulse laser microfabrication: how laser process parameters influence efficiency, throughput and quality." Applied Physics A 120, no. 3 (2015): 847-855.

C- Fikry, Mohamed, Walid Tawfik, and Magdy Omar. "Measurement of the Electron Temperature in a Metallic Copper Using Ultrafast Laser-Induced Breakdown Spectroscopy." Journal of Russian Laser Research 41, no. 5 (2020): 484-490.

D- Fikry, Mohamed, Walid Tawfik, and Magdy M. Omar. "Investigation on the effects of laser parameters on the plasma profile of copper using picosecond laser induced plasma spectroscopy." Optical and Quantum Electronics 52 (2020): 249.

E- Fikry, Mohamed, Walid Tawfik, and Magdy M. Omar. " Controlling the plasma electron number density of copper metal using NIR picosecond laser-induced plasma spectroscopy." Optica Applicata 51 (3) 2021.

F- Li, Yunfan, Xuanqi Guo, Shuai Wang, Yilin Zhao, Chen Zhang, Sheng Liu, Gary J. Cheng, and Feng Liu. "High-precision and high-efficiency copper removal by femtosecond laser in mixed hydrochloric acid‑oxygen atmosphere." Journal of Manufacturing Processes 82 (2022): 152-158.

Response: Thank you for your careful review. The above references are important for the literature review in the introduction. Reference A and B investigated the effect of laser parameters on the removal of Cu from various aspects. Reference C, D and E investigated various aspects of laser-induced Cu plasma. Reference F proposed a Cu removal process by acid atmosphere assisted femtosecond laser. We cited these articles in the introduction (Line 44 ~ 45).

Original: Therefore, laser has been widely used in Cu micromachining, such as surface modification [26-28], nanostructure fabrication [29-31], high-precision removal [32-35] and selective removal [36-38].

Revise: Therefore, laser has been widely used in Cu micromachining, such as surface modification [26-28], nanostructure fabrication [29-31], induction of plasma [32-34], high-precision removal [35-39] and selective removal [40-42].

3- Many technical details should be given; the used laser focusing optics, the generated dc plasma source details, including plasma temperature and density, and how the synchronization is done between the stationary plasma source and the laser?

Response: We deeply appreciate your suggestion. According to the comment, we have added a more detailed interpretation regarding experimental process (Line 75~95).

The laser pulse is focused by a galvanometer, which works by controlling the reflection angle of the incident laser beam between two mirrors with a computer. The two mirrors can be scanned along the X and Y axes respectively to achieve the deflection of the laser beam, so that the laser focus point moves on the Cu film according to the set path.

The plasma generator (SPA-2600, SINDIN Co., Ltd., Dongguan, China) consists of a high voltage excitation power supply, a spray gun, and a control system. Under the excitation of high voltage power supply, the compressed nitrogen with pressure of 0.3MPa is ionized into plasma, and then the plasma is sprayed by the spray gun to act on the copper film. The voltage and frequency of the excitation power supply are 220 V and 40 kHz respectively. The generating power, diameter and temperature of nitrogen plasma flow are 180 W, 5 mm, and 70 ° respectively. During ps-laser irradiation of Cu film, the plasma spray gun is installed on the rack, and the ps-laser ablates Cu in the plasma flow action region to realize the synergy between ps-laser and plasma.

In the revised manuscript, the plasma density is not given. This is because the plasma generator does not display the plasma density. The density of the plasma produced by the generator is controlled by the gas pressure and the output power. In the revised manuscript, the gas pressure and the output power are presented, which is related to the plasma density.

Revised: The processing diagram of nitrogen plasma-assisted ps-laser removal of Cu is shown in Figure 1(a). The wavelength, pulse width, maximum repetition frequency and maximum output power of the ps-laser (TruMicro 5000, TRUMPF Scientific Lasers Co., Ltd., Unterföhring, Germany) are 532 nm, 12 ps, 600 kHz and 100 W, respectively. In the processing for Cu removal, the repetition frequency and defocus of the ps-laser are set to 600 kHz and 0 μm, and the scanning speed (v), scanning pitch (Δd) and spot diameter are 10000 mm/s, 10 μm and 10 μm, respectively. The pulse laser is focused on the sample surface by a galvanometer and scanned on the sample surface according to the path set by the computer.

The plasma generator (SPA-2600, SINDIN Co., Ltd., Dongguan, China) consists of a high voltage excitation power supply, a spray gun and a control system. Under the excitation of high voltage power supply, the compressed nitrogen with pressure of 0.3MPa is ionized into plasma, and then the plasma is sprayed by the spray gun to act on the copper film. The voltage and frequency of the excitation power supply are 220 V and 40 kHz respectively. The generating power, diameter and temperature of nitrogen plasma flow are 180 W, 5 mm, and 70 ° respectively. During ps-laser irradiation of Cu film, the plasma spray gun is installed on the rack, and the ps-laser ablates Cu in the plasma flow action region.

4- The depth profile through the SEM images is not clear. I highly recommend using the TEM images instead.

Response: Thank you for your careful review. We apologize for not describing the depth profile clearer. In this work, we observed surface morphology of the sample by SEM. The removal depth of the Cu film was characterized by an optical profiler (Line 96~99), which is widely used to measure laser ablation depths [1-5]. Specifically, a line scan map of the laser ablation area is obtained by an optical profiler. Then, the removal depth is obtained by calculating the height difference between the untreated area and the ablated area.

Due to the weak penetration of the electron beam, the samples for TEM must be made into ultra-thin samples with a thickness of about 50 nm. However, our samples are multilayer film structures (silicon-based Cu films) and the removal depth is several hundred nanometers. These factors make it very difficult to test the removal depth of our samples by TEM.

In summary, the removal depths in this work were characterized by the optical profiler, and the obtained experimental datum are intuitive and reliable.

References:

  1. Li, Y., et al., High-precision and high-efficiency copper removal by femtosecond laser in mixed hydrochloric acid‑oxygen atmosphere. Journal of Manufacturing Processes, 2022. 82: p. 152-158.
  2. Wang, S., et al., Nanoscale-Precision Removal of Copper in Integrated Circuits Based on a Hybrid Process of Plasma Oxidation and Femtosecond Laser Ablation. Micromachines, 2021. 12(10): p. 1188.
  3. Wang, S., et al., A low-damage copper removal process by femtosecond laser for integrated circuits. Vacuum, 2022. 203: p. 111273.
  4. Sedao, X., et al., Influence of pulse repetition rate on morphology and material removal rate of ultrafast laser ablated metallic surfaces. Optics and Lasers in Engineering, 2019. 116: p. 68-74.
  5. Li, S., et al., Femtosecond laser selective ablation of Cu/Ag double-layer metal films for fabricating high-performance mesh-type transparent conductive electrodes and heaters. Optics Communications, 2021. 483: p. 126661.

Reviewer 2 Report

The high-efficiency removal of copper was realized by nitrogen plasma-assisted picosecond laser. The removal mechanism and simulation were also studied by authors. This research on energy field assisted ultrafast laser processing is very significant. It is recommended to publish this article, but the following modifications are required.

1. Not all lasers have advantages of nanoscale removal accuracy, μm/min removal rate and low mechanical damage (Line 40~41, Introduction). It is necessary to indicate the type of laser.

2. Please supply the crystal orientation, thickness of silicon wafers. Is it doped with other elements? If so, please describe clearly.

3. Should Figure 1(b) be adjusted to the back of the experiment? Because these phenomena need theoretical and experimental proof.

4. It is mentioned that the nitrogen ions in the nitrogen plasma do not react chemically with copper. However, is the role of oxygen taken into account in air processing? Please supply the elemental composition test of the copper film.

5. Is the silicon wafer affected by copper film removal? After all, the copper film is very thin. Please supply SEM/TEM to characterize the interface between copper and silicon wafer. In addition, the material test is too single, so it is suggested to supplement some related tests.

6. It is suggested to use deformed mesh to simulate the removal process of copper film. Is the effect of laser induced plasma on the processing considered in the simulation?

7. Please supply the error bar in Figure 7. Can empirical equations be used to fit the relationship between roughness and removal depth with fluence?

Author Response

Review 2:

The high-efficiency removal of copper was realized by nitrogen plasma-assisted picosecond laser. The removal mechanism and simulation were also studied by authors. This research on energy field assisted ultrafast laser processing is very significant. It is recommended to publish this article, but the following modifications are required.

Response: We feel great thanks for your professional review work on our article. As you are concerned, there are several problems that need to be addressed. According to your professional suggestions, we have made extensive corrections to our previous manuscript, the detailed corrections are listed below.

1. Not all lasers have advantages of nanoscale removal accuracy, μm/min removal rate and low mechanical damage (Line 40~41, Introduction). It is necessary to indicate the type of laser.

Response: Thank you for underlining this deficiency. We agree with this comment and have rewritten this sentence in the revised manuscript. For the sake of rigor, we have changed the term "Laser processing" to "Ultrafast laser processing" (Line 40~41).

Original: Laser processing is a non-contact method with nanoscale removal accuracy, μm/min removal rate, and low mechanical damage [24, 25], which has great application potential in Cu processing of electronic devices.

Revise: The accuracy of laser processing is closely related to the pulse width of the laser used. The laser processing based on ultrafast laser can achieve nanoscale removal accuracy, μm/min removal rate, and low mechanical damage [24, 25], which has great application potential in Cu processing of electronic devices.

2. Please supply the crystal orientation, thickness of silicon wafers. Is it doped with other elements? If so, please describe clearly.

Response: Thank you for your careful review. We use undoped high-purity silicon with crystal orientation (100) and thickness of 400 μm.

Original: Silicon wafers (Lijing Photonics Co., Ltd., Zhejiang, China) are ultrasonically cleaned by using acetone, ethanol, and deionized water in sequence, and then a 1.5 μm thick Cu film is deposited on the wafers by magnetron sputtering.

Revise: High-purity silicon wafers (Lijing Photonics Co., Ltd., Zhejiang, China) with (100) crystal orientation and a thickness of 400 μm are used. The silicon wafers are ultrasonically cleaned by using acetone, ethanol, and deionized water in sequence, and then a 1.5 μm thick Cu film is deposited on the wafers by magnetron sputtering.

3. Should Figure 1(b) be adjusted to the back of the experiment? Because these phenomena need theoretical and experimental proof.

Response: Thank you for your suggestion. We placed Figure 1(b) before the experiment for two reasons. On the one hand, it allows readers to quickly understand the idea of the full text. On the other hand, it deepens the reader's understanding of the subsequent experimental results. It is also worth mentioning that we cited some references as proofs during the discussion of the mechanistic analysis (Line 108~118). The subsequent simulations and experiments can corroborate with the above mechanism analysis.

4. It is mentioned that the nitrogen ions in the nitrogen plasma do not react chemically with copper. However, is the role of oxygen taken into account in air processing? Please supply the elemental composition test of the copper film.

Response: Thank you for your professional advice. We analyzed the elemental composition of the samples by EDS test, as shown in Figure 1. In the figure, the Cu film treated by nitrogen-assisted ps-laser contains a large amount of Cu and a small amount of C and O. This proves that the Cu film does not react with nitrogen after treated by nitrogen-assisted ps-laser. The presence of small amounts of C may be caused by the ablated organic matter remaining on the surface of the Cu film. The presence of element O indicates that a small amount of copper oxide formed by the reaction between Cu and oxygen in the air is deposited on the Cu film during ablation. We have added the EDS analysis to the manuscript on the Section 4.4.

Figure 1. EDS results of the Cu films ablated by the 2.69 J/cm2 ps-laser: (a) without the plasma assistance, (b) with the 30°plasma assistance.

5. Is the silicon wafer affected by copper film removal? After all, the copper film is very thin. Please supply SEM/TEM to characterize the interface between copper and silicon wafer. In addition, the material test is too single, so it is suggested to supplement some related tests.

Response: We think this is an excellent suggestion. The effect of ultrafast laser processing of Cu films on the underlying silicon substrate has been investigated in our previous work (Wang, S., et al. Vacuum. 2022, 203, 111273). To investigate the damage of silicon substrate during fs-laser removal of Cu film, the Cu film near the ablation carter was removed by wet etching to expose the underlying silicon substrate. The 3D profiles of the sample with part of the copper film removed is shown in Figure 5. It can be seen from the figures that there is no trace of ablation on the silicon substrate with the Cu film removed and the profile of the bottom of the ablation crater is as flat as that of the silicon substrate with the Cu film removed, indicating that the fs-laser does not damage the underlying silicon substrate while removing the upper Cu film.

Wang, S., et al. Vacuum. 2022, 203, 111273. Figure 5. (a) Optical image of the sample irradiated by a single-pulse fs-laser with 7.48 J/cm2; (b) 3D morphology and (c) Profile of the sample in which part of the copper film is removed.

In this work, the thickness of the Cu film on the silicon wafer is 1.5 μm, while at most 0.8 μm thick Cu was removed. Therefore, the silicon substrate is not be affected by ps-laser. Because this manuscript focuses on studying the effect of plasma gas flow on ps-laser ablation of Cu film, we have not included a discussion of the machining damage in the manuscript.

In addition, we used XRD patterns to analyze the phases of the Cu film. As shown in Figure 2, the processed Cu film contains a large amount of Cu and a small amount of Cu oxide, but no N phase, which indicates that the nitrogen plasma does not affect the phase composition of the Cu film.

We have added the suggested content to the manuscript on the Section 4.4.

Figure 2. XRD results of the Cu film ablated by the 2.69 J/cm2 ps-laser: (a1) without the plasma assistance, (a2) with the 30°plasma assistance.

Revise: The phases of the Cu films ablated by ps-laser are investigated. Figure 8(a) shows the XRD patterns of the Cu films ablated by the 2.69 J/cm2 ps-laser. In the figure, the peaks of CuO (110), CuO (111), CuO (113) and Cu (111) appear in the sample ablated by ps-laser, which indicates that a small amount of Cu is oxidized to CuO during ps-laser ablation. No peak related to nitrogen element appears in the XRD patterns, which indicates that the nitrogen plasma does not react with Cu. In addition, the pattern of Figure 8(a1) is basically the same as that of Figure 8(a2), indicating that the assistance of nitrogen plasma has little effect on the phase structure of the Cu film. Figure 8(b) shows the elemental compositions of the samples ablated by a 2.69 J/cm2 ps-laser. In the figure, the Cu film ablated by nitrogen-assisted ps-laser contains a large amount of Cu and a small amount of C and O. This proves that the Cu film does not react with nitrogen after ablated by nitrogen-assisted ps-laser, which is consistent with the XRD results. The presence of small amounts of C may be caused by the ablated organic matter remaining on the surface of the Cu film. The presence of element O indicates that a small amount of copper oxide formed by the reaction between Cu and oxygen in the air is deposited on the Cu film during ablation. In addition, the oxygen content of the sample ablated in nitrogen plasma is almost the same as that of the sample ablated in air, further indicating that the assistance of nitrogen plasma has little effect on the phase structure of the Cu film.

Figure 8. (a) XRD results of the Cu film ablated by the 2.69 J/cm2 ps-laser: (a1) without the plasma assistance, (a2) with the 30°plasma assistance. (b) EDS results of the Cu film ablated by the 2.69 J/cm2 ps-laser: (b1) without the plasma assistance, (b2) with the 30°plasma assistance.

6. It is suggested to use deformed mesh to simulate the removal process of copper film. Is the effect of laser induced plasma on the processing considered in the simulation?

Response: Thank you for pointing this out. It is feasible to simulate the Cu removal process by deformed mesh. However, the experimental conditions in this work are complex and the numerical simulation of the removal process is difficult to match well with the experimental values. In the highly simplified two-dimensional model, the theoretical removal depth calculated from the normal grid velocity equation differs significantly from the experimental values. Therefore, we only gave the calculated results for the temperature field in the manuscript. In addition, the analysis of the simulation results in the manuscript is qualitative, and the simulation part is only used as an auxiliary proof of the experimental analysis.

It is undeniable that laser-induced plasma is an important component in the ps-laser-material interaction process. However, the ps-laser in this work is a single-pulse process, and the plasma induced by the single-pulse ps-laser is relatively minute compared to the large amount of nitrogen plasma added directly. Therefore, to simplify the simulation model, we did not take the effect of the laser-induced plasma into account in the simulation. In the future, we will study a more comprehensive simulation model of plasma coupled with ultrafast laser.

7. Please supply the error bar in Figure 7. Can empirical equations be used to fit the relationship between roughness and removal depth with fluence?

Response: We sincerely appreciate the valuable comments. As suggested by the reviewer, we have added the error bar to Figure 7 in the revised manuscript.

It is feasible to fit the data in Figure 7 by empirical equations. However, there are many influencing factors in the laser processing. In our work, the effects of laser fluence and plasma action angle are studied systematically, and many other process parameters have not been studied systematically. Therefore, it is difficult for us to put forward an accurate empirical equation that takes these influencing parameters into account. In the future, we will consider more process parameters (e.g., temperature, flow rate) and conduct a systematic study to obtain a meaningful empirical equation.

Original:

Revise:

Figure 7. Correlation between laser fluence and processing quality of Cu films ablated by nitrogen plasma-assisted ps-lasers: (a) roughness, (b) removal depth.

Round 2

Reviewer 2 Report

It is suggested to publish this article.